# The European Thyroid Imaging and Reporting Data System as a Remedy for the Overdiagnosis and Overtreatment of Thyroid Cancer: Results from the EUROCRINE Surgical Registry

**DOI:** 10.3390/cancers16122237

**Published:** 2024-06-17

**Authors:** Andrzej Rafał Hellmann, Piotr Wiśniewski, Maciej Śledziński, Marco Raffaelli, Jarosław Kobiela, Marcin Barczyński

**Affiliations:** 1General, Endocrine and Transplant Surgery, Medical University of Gdańsk, 80-210 Gdańsk, Poland; maciej.sledzinski@gumed.edu.pl (M.Ś.); kobiela@gumed.edu.pl (J.K.); 2Chair and Department of Endocrinology and Internal Diseases, Medical University of Gdańsk, 80-210 Gdańsk, Poland; piotr.wisniewski@gumed.edu.pl; 3Centro di Ricerca in Chirurgia delle Ghiandole Endocrine e dell’Obesità (C.R.E.O.), Università Cattolica del Sacro Cuore, 00168 Rome, Italy; marco.raffaelli@unicatt.it; 4Division of Endocrine and Metabolic Surgery, Fondazione Policlinico Universitario Agostino Gemelli IRCCS, 00168 Rome, Italy; 5Department of Endocrine Surgery, Faculty of Medicine, Jagiellonian University Medical College, 31-008 Kraków, Poland; marbar@mp.pl

**Keywords:** papillary thyroid carcinoma, EU-TIRADS, FNAB, thyroid nodules

## Abstract

**Simple Summary:**

The European Thyroid Imaging and Reporting Data System (EU-TIRADS) aids clinicians in determining the necessity of fine-needle aspiration biopsy (FNAB) for thyroid nodules suspected of malignancy. This study evaluated the effectiveness of EU-TIRADS criteria in real-world surgical settings. Results indicated that the EU-TIRADS demonstrates reliable diagnostic accuracy in identifying thyroid cancer, facilitating improved clinical decision-making regarding patient management. Enhanced adherence to EU-TIRADS guidelines can mitigate the risk of an overdiagnosis and overtreatment, particularly for low-risk thyroid cancer cases, thereby optimizing patient care.

**Abstract:**

Background: The European Thyroid Imaging and Reporting Data System (EU-TIRADS) aims to reduce the overdiagnosis of thyroid cancer (TC) by guiding the selection of nodules for fine-needle aspiration biopsy (FNAB). This study sought to validate EU-TIRADS nodule selection criteria using data from EUROCRINE, an extensive international endocrine surgery registry. Method: We reviewed indications for FNAB among patients with TC compared to those with benign disease who underwent surgery between March 2020 and March 2022, considering preoperative EU-TIRADS scores and dominant nodule size (FNAB is recommended in Category 5 (˃10 mm or ˂10 mm with suspicious lymph nodes), 4 (˃15 mm), and 3 (˃20 mm)). Patients were categorized into three risk groups: minimal risk (patients with papillary microcarcinoma), high risk (patients with pT3b stage or higher, pN1b, or pM1), and low–moderate risk (all other patients). We conducted a Receiver Operating Characteristic (ROC) analysis to assess the diagnostic accuracy of the EU-TIRADS. Results: We analyzed 32,008 operations. Approximately 68% of the surgical records included EU-TIRADS classifications. The EU-TIRADS exhibited diagnostic accuracy across high-volume sites, with a median ROC Area Under the ROC Curve (AUC) of 0.752, indicating its effectiveness in identifying malignancy. Among the cases, 7907 patients had TC. Notably, 55% of patients with TC underwent FNAB despite not initially meeting the EU-TIRADS criteria. These patients were distributed across the minimal- (58%), low–moderate- (36%), and high-risk (5.8%) categories. Of the patients with TC recommended for FNAB, 78% were deemed low–moderate risk, 21% high risk, and only 0.7% minimal risk. Conclusion: The EU-TIRADS offers effective preoperative malignancy risk stratification. Promoting the proper use of the EU-TIRADS in clinical practice is essential to mitigate the overdiagnosis and overtreatment of low-risk TC.

## 1. Introduction

The most common endocrine disorder is thyroid nodules, which are diagnosed at an ever-increasing frequency. Around 19–68% of adults have been detected with at least one nodule. Of those discovered thyroid nodules, approximately 7–15% may progress to thyroid carcinoma [1]. The detectability of thyroid cancer has significantly increased over the past decade, and this change is mainly attributed to the widespread use of ultrasonography, as confirmed by various studies [2,3,4]. High-resolution ultrasound (US) is the most sensitive imaging technique for detecting and describing the number, size, and topography of thyroid nodules with respect to glandular architecture. Additionally, the common use of US results in a rapid increase in the number of lesions of the thyroid gland diagnosed as clinically irrelevant. Differentiating malignant nodules from benign nodules remains a challenge.

The most accurate method for evaluating thyroid nodules and selecting patients for thyroid surgery is fine-needle aspiration biopsy (FNAB). Several thyroid nodule risk classification systems focusing on ultrasound (US) features have been published. Some emphasize simple US patterns, while others rely on multiple US features to categorize thyroid nodules. However, the classification and selection of thyroid nodules for FNAB mostly depends on radiologists’ individual experience and is of low reproducibility [5]. This situation leads to an overdiagnosis and numerous unnecessary FNABs, which is one of the main reasons for proposing a US classification for the assessment of the risk of malignancy of thyroid nodules, called the Thyroid Imaging Reporting and Data System (TIRADS). Since it was first published by Horvath in 2009, several modifications have been developed across the world. In 2015, Grant et al. published a thyroid ultrasound reporting lexicon in which all thyroid nodules were classified on the basis of TIRADS categories, which, in turn, not only defined their risk of malignancy but offered evidence-based recommendations to manage thyroid nodules based on their size and sonographic features [6]. After the first Korean version of the TIRADS system by Kwak et al. [7], Shin et al. (2016) subsequently proposed a revised Korean Society of Thyroid Radiology (KSThR) consensus statement with recommendations in which specific sonographic features were used to stratify the risk of thyroid nodules into four categories [8]. According to the published literature, the Korean TIRADS has been successfully used for the US evaluation of thyroid nodules in order to stratify the need for these nodules to undergo FNA. The 2015 ATA guideline includes a detailed description of sonographic features, categorizing thyroid nodules that utilize one of the described patterns. The most suspicious US features include margins, microcalcifications, “taller-than-wide” shape, rim calcifications, and evidence of extrathyroidal extension. Specifically, the ATA defined and identified five categories: (1) benign (ROM < 1%); (2) very low suspicion (ROM < 3% in lesions ≥ 20 mm); (3) low suspicion (ROM 5–10% in lesions ≥ 15 mm); (4) intermediate suspicion (ROM 10–20% in lesions ≥ 10 mm); and (5) high suspicion (ROM 70–90% in lesions ≥ 10 mm). In 2017, the European Thyroid Association (ETA) published the novel European Thyroid Imaging and Reporting Data System (EU-TIRADS). This grading system evaluates the risk of potential malignancy using a pattern system based on the composition of the nodule, echogenicity, shape, margin, and echogenic foci, with higher-scoring nodules being more likely to be malignant. According to this system, two-to-five categories are proposed: EU-TIRADS 2 (benign category with risk of malignancy (ROM) close to 0%), EU-TIRADS 3 (low-risk category with ROM of 2–4%), EU-TIRADS 4 (intermediate-risk category with ROM of 6–17%), and EU-TIRADS 5 (high-risk category with ROM of 26–87%) [9].

Studies report that the effective implementation of TIRADS could eliminate over half of unnecessary FNABs, which, apart from increased costs (ranging from approximately USD 500 to USD 3000, including molecular tests) [10], carries an increased risk of complications and stress for the patient [11]. While TIRADS has demonstrated effectiveness, its integration into national guidelines remains limited. Only recently has the Polish Endocrine Society adopted the EU-TIRADS in its guidelines [12]. In an examination of the knowledge and use of TIRADS in Italy, Mauri et al. found that only 53.6% of radiologist respondents were familiar with the classification [13]. Since the EU-TIRADS was first described, a few validation studies have been published. The most important report by Trimboli et al. showed that the EU-TIRADS is an excellent tool for discriminating high-ROM thyroid nodules from those with a low or very low likelihood of being cancer [14]. Another multicenter study performed in a previously iodine-deficient region showed similar results [15]. However, the limitation of both studies was their retrospective character and the relatively small number of patients included. The aim of the present project was to validate EU-TIRADS nodule selection criteria using data from EUROCRINE^®^, a large prospective international endocrine surgery registry.

## 2. Materials and Methods

### 2.1. Study Population Inclusion and Exclusion Criteria

The present study was approved by the Independent Bioethics Committee for Scientific Research of the Medical University of Gdansk (NKBBN/355/2021). The data were obtained from the EUROCRINE^®^ registry for endocrine surgery.

The European registry EUROCRINE^®^ was established in 2015 for hospitals with a focus on endocrine surgery. The objectives of the EUROCRINE^®^ registry is to reduce morbidity and mortality from diseases of the endocrine system through an international comparison of applied therapy strategies [16]. The registry offers the possibility of recording indications, preoperative diagnoses, perioperative management, and extensive postoperative follow-up examinations. Data integrity and validity are checked through audits and internal algorithms for plausibility checks.

Inclusion Criteria:Patients with thyroid nodules who underwent surgery between March 2020 and March 2022.Availability of preoperative EU-TIRADS scores and dominant nodule size.

Exclusion Criteria:Patients without recorded EU-TIRADS classification.Incomplete data on primary and secondary histological diagnoses.

The data were extracted in April 2022; the flow chart of the study population selection is shown in Figure 1.

### 2.2. Study Outcomes

In the EUROCRINE data set, subjects were assigned EU-TIRADS classes based on the most suspicious lesion. The assessment of EU-TIRADS diagnostic performance in detecting malignant lesions was determined solely using the main histological diagnosis, with two secondary histological diagnoses provided but not considered for malignancy classification.

To study the use of the EU-TIRADS based on EUROCRINE^®^ participants, we calculated the percentage of operations with the EU-TIRADS reported in the whole data set and within each of the reporting institutions.

The rate of malignancy for each EU-TIRADS class was calculated to determine the diagnostic performance of the EU-TIRADS in discriminating benign from malignant thyroid nodules. A nodule was considered malignant if any of the following diagnoses were reported as the primary histopathological diagnosis: papillary cancer, anaplastic cancer, follicular cancer, medullary cancer, Hürthle cell (oncocytic) carcinoma, lymphoma, poorly differentiated TC, or a metastatic tumor. Noninvasive follicular thyroid neoplasms with papillary-like nuclear features (NIFTP) were also included in the malignant category.

Among patients with malignant tumors, we distinguished three clinical risk profiles based on histological type and TNM stage. The eighth edition of the TNM staging system was used [17]. Patients with a diagnosis of NIFTP or papillary TC pT1a pN0/Nx M0/Mx were assigned to the minimal-risk category. Patients with a tumor stage of pT3b or higher, pN1b, or M1 were assigned to the high-risk category. The remaining patients constituted the low–moderate-risk category.

We also investigated the number of malignant tumors with and without FNAB indications according to EU-TIRADS guidelines. EU-TIRADS guidelines recommend FNAB for nodules in Category 5 (nodules ˃ 10 or ˂ 10 mm when suspicious lymph nodes are present), Category 4 (nodules ˃ 15 mm), and Category 3 (nodules ˃ 20 mm) [9]. FNAB is not indicated for nodules in Category 2. In the EUROCRINE database, the size of the nodule is provided for malignant, but not benign, tumors.

### 2.3. Statistical Methods

Continuous parameters reported in the manuscript are summarized as the median and interquartile ranges (IQRs), unless otherwise specified. Categorical parameters are expressed as counts and percentages. For the analysis of EU-TIRADS diagnostic accuracy, Class 4–5 results were considered “test-positive”, while Classes 1–3 were considered “test-negative”. As the gold standard, a histopathological diagnosis of thyroid malignancy was used. ROC AUC, accuracy, sensitivity, specificity, positive and negative likelihood ratios, and positive and negative predictive values were calculated for each EU-TIRADS class. The data processing and analysis were performed using the R statistical environment version 4.2.1 [18] and RStudio IDE release 2022.07.1+554 [19].

## 3. Results

A total of 32,008 thyroid operations performed at 113 sites in 10 countries were registered between 1 March 2020 and 31 March 2022. A median of 83 (IQR: 43–284, range: 1–7506) procedures were entered per site. Overall, 24 sites (21.2%) reported fewer than 40 procedures.

In all, 24,950 thyroid operations (77.9%) were performed in females and 7058 (22.1%) in males. The median patient age was 51 (IQR: 39–62) years. The main indications for surgery were the exclusion of malignancy in 12,914 (40.3%) cases, compression symptoms in 6643 (20.8%), malignancy in 5657 (17.7%), and thyrotoxicosis in 5397 (16.9%). Total thyroidectomy was performed in 15,419 operations (48.2%) and unilateral lobectomy was performed in 13,742 (42.9%). Lymph node surgery was performed in 6299 operations (19.7%). Malignant neoplasm was the main histopathological diagnosis in 10,254 operations (32.0%). See Table 1 for more details on demographics, indications for surgery, types of surgery, and main histopathological diagnoses.

Incidental malignant neoplasms were found in 723 of 7347 (9.8%) cases with a non-oncological main indication for surgery (thyrotoxicosis, compression symptoms, recurrent cyst, others). Of the incidental neoplasms, 423 (59%) were in the minimal-, 282 (39%) in the low–intermediate, and 18 (2%) in the high-clinical-risk category.

### 3.1. EU-TIRADS Reporting Rate

The EU-TIRADS classification was reported for 21,925 of the 32,008 operations (67.8%). Class 1 was reported in 1289 (5.9%), Class 2 in 2504 (11%), Class 3 in 4428 (20%), Class 4 in 7126 (33%), and Class 5 in 6578 (30%) of the operations.

Among the 89 sites that registered at least 40 operations, the proportion of submissions containing the EU-TIRADS classification was 0.63 (0.25–0.87). There was no correlation between the number of reported operations and the proportion of cases submitted with the EU-TIRADS classification.

The percentage of submissions containing the EU-TIRADS classification varied depending on the indication for surgery. For the indications ‘Malignancy,’ ‘Excluding malignancy,’ ‘Compression symptom,’ and ‘Thyrotoxicosis,’ the percentages were 80%, 73%, 68%, and 54%, respectively. In the case of the remaining indications, 43% were assigned an EU-TIRADS class.

### 3.2. EU-TIRADS Diagnostic Performance

Of the 20,762 operations included in the analysis, 7907 (38%) had a diagnosis of a malignant thyroid neoplasm. A total of 1229 nodules (6.0%) were classified as EU-TIRADS Class 1, 2412 (12%) as Class 2, 4133 (20%) as Class 3, 6831 (33%) as Class 4, and 6157 (30%) as Class 5.

The prevalence of malignant neoplasms was lowest in nodules classified as EU-TIRADS Class 1 (7.7%) and increased in each subsequent class, reaching 10% in Class 2, 19% in Class 3, 33% in Class 4, and 73% in Class 5. For EU-TIRADS Class 5 nodules, the accuracy, sensitivity, specificity, positive predictive value, and negative predictive value in differentiating nonmalignant from malignant nodules were 0.758, 0.571, 0.872, 0.733, and 0.768, respectively (Appendix A). For a malignant neoplasm, the odds of being classified as EU-TIRADS Class 5 were 9.1 times higher (odds ratio, OR = 9.1, 95% CI: 85–9.7) than for nonmalignant nodules. The ROC curve analysis (Figure 2) showed that the EU-TIRADS had an acceptable overall ability to discriminate malignant from nonmalignant thyroid conditions, with an AUC of 0.78 (95% CI from 0.77 to 0.78). The distribution of thyroid cytology results within each EU-TIRADS category reveals varying patterns, with EU-TIRADS 4 and 5 predominantly associated with higher-risk cytological findings (Table 2).

Histopathological types of neoplasms diagnosed in each EU-TIRADS class are summarized in Table 2. Papillary and medullary cancers were found mostly in Class 5, while follicular-patterned neoplasms were predominantly in Class 4.

Of the 7907 histopathologically diagnosed malignant neoplasms, 6766 (86%) were papillary TCs, 5762 (73%) were stage pT1, 2201 (28%) were lymph-node-positive (pN1), and 96 (1.2%) were diagnosed with metastases (pM1). Overall, 2607 (33%) cases were in the minimal-clinical-risk category, 4297 (54.3%) were in the low–moderate-risk category, and 1003 (12.7%) were in the high-risk category. More clinicopathological details are presented in Table 3.

### 3.3. Recommended and Not Recommended FNABs

According to EU-TIRADS guidelines, indications for FNAB were present in 3517 (44.5%) of 7907 patients diagnosed with malignant thyroid tumors. Those patients for whom FNABs were not recommended were slightly older (median age: 49 vs. 47 years), with a greater prevalence among females (83% vs. 76%), and were treated more frequently with lobectomy (42% vs. 34%) compared to patients with FNAB indicated by the EU-TIRADS. The pathological findings of malignant tumors in both subgroups are shown in Table 3. Patients for whom FNA was not recommended had fewer tumors with gross extrathyroidal invasion (stage pT3b or higher, 0.9% vs. 7.3%), and regional lymph node involvement was less prevalent (pN1, 19.8% vs. 38%). The histological type was more commonly papillary thyroid carcinoma (PTC) in patients for whom FNA was not recommended (93% vs. 81%). Clear differences in terms of clinical risk were noted between the two subgroups. Minimal-risk tumors were found in less than 1% of cases with recommended FNAB vs. 61% of cases where FNAB was not recommended. Table 3 provides more details on this subject.

## 4. Discussion

The EU-TIRADS classification system is a US-based risk stratification method for thyroid nodules. Its purpose is to reduce the overdiagnosis and overtreatment of TC by guiding the selection of nodules for FNAB. A worldwide increase in the incidence of TC has been observed, primarily driven by the overdiagnosis of small papillary carcinomas [20]. The authors of this important report stated that improved diagnostic techniques and increased medical surveillance have led to the detection of asymptomatic thyroid tumors that would not have caused symptoms or death if left untreated. The findings highlight the need for caution in systematic screening for TC and advocate for watchful waiting approaches for low-risk cases rather than immediate surgery and overtreatment. Since the introduction of the novel EU-TIRADS classification system, several studies have been conducted to evaluate its diagnostic performance in clinical practice [14,15,21,22,23,24,25].

Recent research on this subject has been summed up in a meta-analysis by Castellana et al. [26], and the results have varied among different centers across Europe, highlighting the need for a large European multicenter study to establish the diagnostic value of the EU-TIRADS and verify its applicability across different centers. To our knowledge, this represents the first multicenter analysis conducted in European countries where uniform data entry procedures were maintained across all participating centers.

This study aimed to validate the effectiveness of the EU-TIRADS in reducing the overdiagnosis and overtreatment of TC. The authors reviewed the indications for FNAB in patients who underwent surgery and compared the preoperatively reported EU-TIRADS scores and the size of the dominant lesion with the histopathological diagnosis of TC.

This study analyzed over 32,000 operations from more than 120 European surgical departments. Among the cases analyzed, 68% had an EU-TIRADS classification available. This result is an improvement compared to the previously reported rate of 53.6% by Mauri et al. for Italy [13], as well as our own yet unpublished results from a Polish national survey conducted in 2022, which showed a rate of 43%. These findings suggest that the implementation of the EU-TIRADS is gradually increasing, although there is still a significant percentage of cases in which it is not used.

This study found that the EU-TIRADS scale had a moderate-to-good ability to predict malignancy, with an AUC ranging from 0.626 to 0.895 across different high-volume sites. The best performance was for EU-TIRADS Class 5 nodules when the accuracy, sensitivity, specificity, positive predictive value, and negative predictive value in differentiating nonmalignant from malignant nodules were 0.758, 0.571, 0.872, 0.733, and 0.768, respectively (Appendix A). This is concordant with Trimboli et al., in which EU-TIRADS 5 had a significantly higher cancer rate than the other summed categories (77%; *p* < 0.0001) with an OR of 84.7. However, these results differ significantly from those published by Dobruch-Sobczak et al. [15], in which the accuracy and sensitivity for EU-TIRADS 5 were 45.5 and 66.4, respectively. The incidence of malignant neoplasms in the EU-TIRADS 5 category (Table 2) was 73%, which falls within the ETA risk range (26–87%) but is closer to the upper limit. In the EU-TIRADS 4 category, the frequency in our analysis was 33%, which is significantly higher than the risk according to the ETA (6–17%). Similarly, Categories 3 and 2 exhibited higher frequencies than the expected risk (3, 19% vs. 2–4%; 2, 10% vs. close to 0; and 1, 8% vs. none). These results raise concerns, particularly regarding lower-risk groups.

It is critical to mention that this study found that a significant proportion of patients with a histopathological diagnosis of TC did not meet the EU-TIRADS criteria for FNAB recommendation but underwent the procedure as part of the workup that led to a referral for surgery. Similar findings were published by Grani et al. [21]. In our study, among patients with a histopathological diagnosis of TC, 55% of cases did not have an indication for FNAB based on the EU-TIRADS classification, but it was nevertheless performed as part of the workup that resulted in referral for surgery.

These patients were carefully categorized into different risk groups based on their diagnostic results. Remarkably, 58% of the patients were found to be in the minimal-risk category, 36% fell into the low–moderate-risk category, and a mere 5.8% were classified as high risk. Particularly striking is the fact that minimal-risk tumors were identified in less than 1% of the cases that underwent the recommended FNAB. This outcome underscores the importance of our findings and highlights why the implementation of the EU-TIRADS is an effective approach for reducing overtreatment. To put the impact into perspective, had all participants in the EUROCRINE^®^ study adhered to the EU-TIRADS guidelines for biopsy, a staggering 4145 unnecessary surgical operations could have been avoided. Notably, among these operations, high-risk patients accounted for less than 6%. This underlines the potential for the EU-TIRADS approach to prevent unnecessary interventions and to ensure more appropriate and targeted care for patients. This finding indirectly helps explain the rapid worldwide increase in the incidence of papillary thyroid carcinoma [27]. PTC is characterized by a very low mortality risk of less than 1% in the 20 years following thyroid surgery [28].

This was a pragmatic study. The ultrasound assessment of the dominant thyroid nodule(s) and categorization into the EU-TIRADS scale was routinely undertaken by different specialists including radiologists, endocrinologists, and surgeons, according to local policies, reflecting a variety of contemporary clinical routines in different European countries. This diversity in US providers demonstrated the practical applicability of the TIRADS scale in everyday clinical practice. However, this aspect also introduced a limitation to our research. We had no control over which patients were examined by a particular group of US specialists; therefore, we were unable to investigate its influence.

While interpreting the observed malignancy rates in our study, it is imperative to acknowledge the distinctive nature of our cohort. The data were obtained from a surgical registry, which means that it focused only on patients who underwent surgical treatment and had available histopathological results. This study did not fully evaluate the accuracy of the EU-TIRADS scale itself. It is crucial to recognize that our findings may not be directly extrapolated to the general population undergoing thyroid evaluations. However, a similar approach and findings were reported previously by Inabnet et al. [29] and in the available meta-analysis by Castellana et al. [26]. The results effectively elucidate the specificity of the EU-TIRADS scale, although the sensitivity aspect remains incompletely evaluated. Within the research conducted by Xu et al. [25], the methodology offers a more refined approach to assessing the sensitivity of the TIRADS scale. This study encompassed a cohort of patients exhibiting a Bethesda II result in FNAB yet they exhibited no evidence of tumor progression within the subsequent year. It is noteworthy, however, that beyond this study, there remained individuals who did not meet the criteria for FNAB and were subjected to extended follow-up exceeding one year.

Additionally, the following limitations of this study should also be recognized: limited core data available for assessment in the thyroid module of the EUROCRINE register (for example, details on preoperative serum calcitonin levels or thyroid scintigraphy are not reported to the register if the preoperative work-up indicates benign thyroid disease) [30], the potential for interobserver variability in the application of the EU-TIRADS and FNA cytology across different centers, and variability in expertise between high-volume and low-volume centers, which can influence the reliability and accuracy of the data collected [31].

Finally, the study group may not represent the usual indications for thyroid surgery, as a significant percentage of patients had oncological indications, possibly due to the impact of the COVID-19 pandemic, which caused the postponement of non-oncological procedures. According to Polish statistical data, in 2019, before the COVID-19 pandemic, the percentage of surgically treated patients diagnosed with TC was 24.4%, while it was 36.8% in 2020 [32]. Recently, Medas et al. proved that, during the COVID-19 pandemic period, there was an increased occurrence of aggressive thyroid tumors [33].

## 5. Conclusions

In conclusion, this study provides clear evidence of the effectiveness of the EU-TIRADS classification for preoperative malignancy risk stratification. While the prevalence of EU-TIRADS classification has been increasing for the past few years, this study has illustrated that further efforts should be made to disseminate proper use of the EU-TIRADS in clinical practice, since this could limit the overdiagnosis and overtreatment of low-risk TC, ultimately improving patient care.

## Figures and Tables

**Figure 1 cancers-16-02237-f001:**
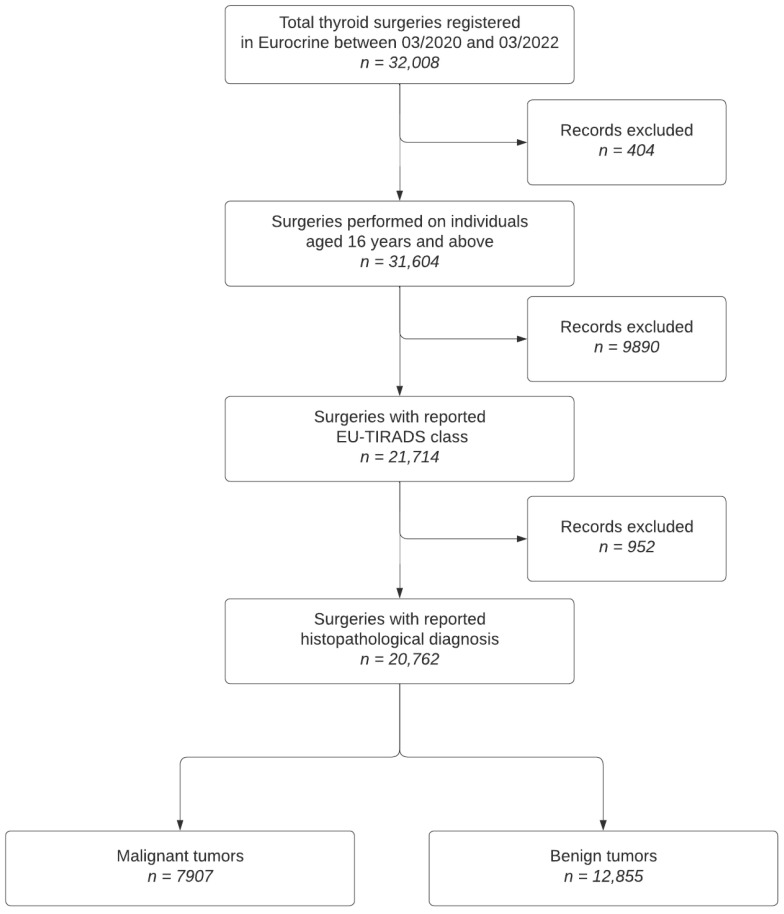
Study flow chart.

**Figure 2 cancers-16-02237-f002:**
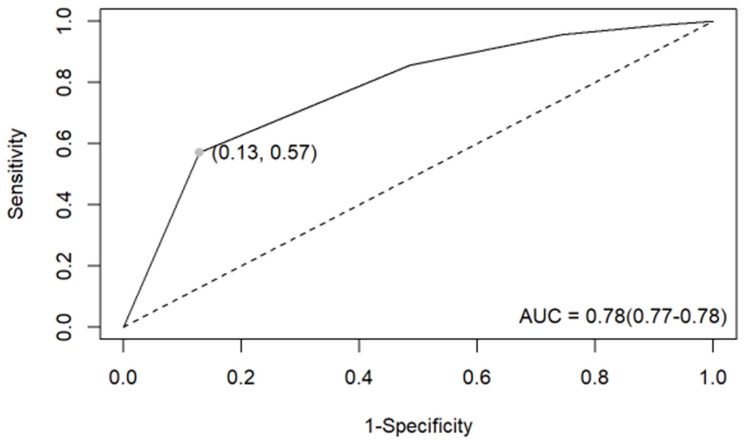
ROC curve for EU-TIRADS in discrimination between malignant and nonmalignant thyroid conditions.

**Table 1 cancers-16-02237-t001:** Patient demographics, main indications for operation, types of operations, and main histopathological diagnoses.

Characteristics	Complete Data Set	Analytic Data Set
Total operations	32,008	20,762
Total patients	31,703	20,691
Females/males	24,950/7058	16,508/4254
Median age, years (min.–max.)	51 (0–106)	51 (16–106)
Indication for surgery:		
Excluding malignancy	12,914 (40%)	8842 (43%)
Compression symptoms	6643 (21%)	4318 (21%)
Malignancy	5657 (18%)	4235 (20%)
Thyrotoxicosis	5397 (17%)	2817 (14%)
Other	1397 (4.4%)	550 (2.6%)
Type of surgery:		
Thyroidectomy	15,419 (48%)	9465 (46%)
Unilateral lobectomy	13,742 (43%)	9647 (46%)
Other	2847 (9%)	1650 (8%)
Lymph node operation:		
None	25,709 (80%)	16,218 (78%)
CLND ^a^	3746 (12%)	3009 (14%)
CLND ^a^ + LLND ^b^	1344 (4.2%)	984 (4.7%)
Other	209 (3.8%)	551 (2.7%)
Histopathological main diagnosis:		
Nodular goiter	9563 (30%)	5346 (26%)
Papillary cancer	8581 (27%)	6766 (33%)
Follicular adenoma	5527 (17%)	4513 (22%)
Graves’ disease	2827 (8.8%)	1569 (7.6%)
Oncocytic adenoma	749 (2.3%)	589 (2.8%)
Follicular cancer	593 (1.9%)	395 (1.9%)
Medullary cancer	423 (1.3%)	319 (1.5%)
Lymphocytic thyroiditis	522 (1.6%)	280 (1.3%)
Oncocytic carcinoma	241 (0.8%)	183 (0.9%)
NIFTP	217 (0.7%)	133 (0.6%)
Other malignancies	199 (0.6%)	111 (0.5%)
Total malignant neoplasms	10,254 (32.0%)	7907 (38.1%)

Values are counts (% of column total). ^a^ Central lymph node dissection, ^b^ lateral lymph node dissection.

**Table 2 cancers-16-02237-t002:** Histopathological diagnoses in each EU-TIRADS class.

Histopathological Type	EU-TIRADS 1 (n = 93)	EU-TIRADS 2 (n = 243)	EU-TIRADS 3 (n = 796)	EU-TIRADS 4 (n = 2259)	EU-TIRADS 5 (n = 4516)
Papillary	79 (1%)	205 (3%)	592 (9%)	1816 (27%)	4074 (60%)
Follicular	3 (1%)	18 (5%)	86 (22%)	207 (52%)	81 (20%)
Medullary	1 (0%)	6 (2%)	35 (11%)	54 (17%)	223 (70%)
Oxyphylic	2 (1%)	4 (2%)	30 (16%)	101 (55%)	46 (25%)
NIFTP	4 (3%)	9 (7%)	48 (36%)	52 (39%)	20 (15%)
Other	4 (4%)	1 (1%)	5 (4%)	29 (26%)	72 (65%)

Values are counts (% of row total).

**Table 3 cancers-16-02237-t003:** Histopathological findings in malignant thyroid tumors (TNM, eighth ed.): comparison of subgroups with a “no biopsy” and “biopsy” recommendation according to EU-TIRADS.

Characteristics	Overall (N = 7907)	FNA Not Recommended (n = 4145 ^1^)	FNA Recommended (n = 3517 ^1^)
Histological type			
Papillary	6766 (86%)	3850 (93%)	2836 (81%)
Follicular	395 (5.0%)	91 (2.2%)	294 (8.4%)
Medullary	319 (4.0%)	154 (3.7%)	158 (4.5%)
Oxyphylic	183 (2.3%)	37 (0.9%)	144 (4.1%)
Other	111 (1.4%)	13 (0.3%)	84 (2.4%)
Primary tumor			
pT1a	3502 (44%)	3407 (82%)	42 (1.2%)
pT1b	2260 (29%)	607 (15%)	1629 (46%)
pT2	1267 (16%)	57 (1.4%)	1198 (34%)
pT3a	425 (5.4%)	34 (0.8%)	383 (11%)
pT3b or higher	301 (3.8%)	38 (0.9%)	258 (7.3%)
pTx	152 (1.9%)	2 (<0.1%)	7 (0.2%)
Regional lymph node			
pN0	3497 (44%)	2076 (50%)	1385 (39%)
pN1a	1414 (18%)	636 (15%)	768 (22%)
pN1b	787 (10.0%)	197 (4.8%)	574 (16%)
pNx	2209 (28%)	1236 (30%)	790 (22%)
Distant metastasis			
pM0	6535 (83%)	3521 (85%)	2967 (84%)
pM1	96 (1.2%)	12 (0.3%)	80 (2.3%)
pMx	1276 (16%)	612 (15%)	470 (13%)
Clinical risk ^2^			
Minimal	2607 (33.0%)	2541 (61.3%)	24 (0.7%)
Low–moderate	4297 (54.3%)	1369 (33.0%)	2743 (78.0%)
High	1003 (12.7%)	235 (5.7%)	750 (21.3%)

Values are counts (% of column total). ^1^ Does not add to 7907 due to 245 cases with missing data. ^2^ Minimal = NIFTP or papillary TC pT1a pN0/Nx M0/Mx; high = stage pT3b or higher or pN1b or M1; low–moderate = remaining.

## Data Availability

Anonymized data will be made available after reasonable request to hellmannandrzej@gmail.com.

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
