# Peer review of "The European Thyroid Imaging and Reporting Data System as a Remedy for the Overdiagnosis and Overtreatment of Thyroid Cancer: Results from the EUROCRINE Surgical Registry"

_cancers, 2024, doi:10.3390/cancers16122237_

Round 1
Reviewer 1 Report
Comments and Suggestions for Authors
The good feature of this paper is represented by the high number of thyroid surgeries. It is well-written and it could improve the evidence about EU-TIRADS.
Some limitations have to be added in the Discussion: the selection bias also caused by the inaccurate selection of nodules submitted to surgery (i.e., details on serum calcitonin, thyroid scintigraphy are not reported (for the latter issue i recommend to cite "Reappraising the role of thyroid scintigraphy in the era of TIRADS: A clinically-oriented viewpoint. Endocrine. Published online April 16, 2024. doi:10.1007/s12020-024-03825-0"); the interobserver variability in the use of EU-TIRADS; the different result of FNC according to the center (for the latter issue, i recommend to cite "Repeat thyroid FNAC: Inter-observer agreement among high- and low-volume centers in Naples metropolitan area and correlation with the EU-TIRADS. Front Endocrinol (Lausanne). 2022;13:1001728").
Author Response
Dear Reviewer,
Thank you for your valuable feedback and suggestions. We appreciate your acknowledgment of the high number of thyroid surgeries and the well-written nature of our paper. We agree that recognizing the limitations of our study is important.
In response to your comments, we have added the following limitations to the Discussion section:
"The following limitations of this study should also be recognized: limited core data available for assessment in the thyroid module of the Eurocrine register (for example, details on preoperative serum calcitonin levels or thyroid scintigraphy are not reported to the register if the preoperative work-up indicates benign thyroid disease), the potential for interobserver variability in the appli-cation of EU-TIRADS and FNA cytology across different centers, and variability in expertise between high-volume and low-volume centers which can influence the reliability and accuracy of the data collected."
Additionally, we have included the references you suggested:
- "Reappraising the role of thyroid scintigraphy in the era of TIRADS: A clinically-oriented viewpoint. Endocrine. Published online April 16, 2024. doi:10.1007/s12020-024-03825-0"
- "Repeat thyroid FNAC: Inter-observer agreement among high- and low-volume centers in Naples metropolitan area and correlation with the EU-TIRADS. Front Endocrinol (Lausanne). 2022;13:1001728"
We hope these revisions address your concerns and improve the manuscript.
Best regards,
Reviewer 2 Report
Comments and Suggestions for Authors
The authors evaluated the effectiveness of European Thyroid Imaging and Reporting Data System (EU-TIRADS) criteria in real world surgical settings. Results indicated that EU-TIRADS demonstrates reliable diagnostic accuracy in identifying thyroid cancer, facilitating improved clinical decision-making regarding patient management. Enhanced adherence to EU-TIRADS guidelines can mitigate the risk of overdiagnosis and overtreatment, particularly for low-risk thyroid cancer cases, thereby optimizing patient care.
Comments and suggestions:
- In the background, provide a more comprehensive overview of previous studies on risk stratification methods and their limitations about thyroid nodules, which will help highlight the novelty of the EU-TIRADS classification system. ​
- Clarified the inclusion and exclusion criteria for patient selection. Additionally, to enhance the validity and generalizability of their results. the authors should discuss any potential biases or limitations associated with the use of surgical registry data.
In summary, I recommend accepting after minor revision.
Author Response
Dear Reviewer,
Thank you for your valuable comments and suggestions. We have made the following revisions to the manuscript:
We have provided a more comprehensive overview of previous studies on risk stratification methods for thyroid nodules and their limitations. This addition helps to highlight the EU-TIRADS classification system.
We have clarified the inclusion and exclusion criteria for patient selection and discussed potential biases and limitations associated with the use of surgical registry data.
All changes have been incorporated into the text of the manuscript and are highlighted in red.
We believe these revisions address the reviewer's concerns and improve the overall quality of the manuscript.
Reviewer 3 Report
Comments and Suggestions for Authors
The aim of the article by A.F.Hellmann et al is to show the effectiveness of EU-TIRADS classification of thyroid nodules in preoperative malignancy risk of thyroid cancer (TC) and stratification of patients to optimize their medical treatment. On the basis of the analysis of EUROCRINE database in combination with the EU-TIRADS guidelines the authors convincingly showed that its proper use in clinical practice could limit overdiagnosis and overtreatment (in particular, significantly reduce the use of biopsies FNAB) for low-risk TC thus improving patient care.
At the same time there are some comments to material presentation. It was obviously unsuccessful to use a comma when separating the digits of large numbers. As a result for example classification of nodules by EU-TIRADS (line 183) looks confusing: Class 1, 2,412 (12%), Class 2, 4,133 (20%) as Class 3, 6,831 (33%) as Class 4, 184 and 6,157 (30%) as Class 5. Only after some concentration of consciousness do you begin to understand that it should really be like this: Class 1 - 2 412 (12%), Class 5 – 6 157(30%) and so on. Also the contents of Table 1 raise some questions. It is not clear, for example, what “Unique patients”(line 2 in the Table) means. In the text (line 152) medium age of patients is indicated as 51 (IQR 39-62) years while in the Table1 the same medium age 51 is indicated as (0-106) or (16-106)? While the origin of the value 20 762 – the number of the analyzed operations is illustrated by Fig.1, the origin of the number of the unique patients 31 703 remains obscure.
All these comments are not fundamental, but interfere with the perception of the material, so it is better to correct them and provide additional clarification. The article is interesting, provide new important information and can be published in Cancer.
Author Response
Your feedback on the presentation of the material is invaluable for enhancing the clarity and comprehensibility of our findings.
- It was obviously unsuccessful to use a comma when separating the digits of large numbers. As a result for example classification of nodules by EU-TIRADS (line 183) looks confusing: Class 1, 2,412 (12%), Class 2, 4,133 (20%) as Class 3, 6,831 (33%) as Class 4, 184 and 6,157 (30%) as Class 5. Only after some concentration of consciousness do you begin to understand that it should really be like this: Class 1 - 2 412 (12%), Class 5 – 6 157(30%) and so on.
Answer: The thousands separator (coma) has been removed throughout the manuscript
- Also the contents of Table 1 raise some questions. It is not clear, for example, what “Unique patients”(line 2 in the Table) means.
Answer: the ambiguous term 'unique patients' has been replaced with 'Total patients'
- In the text (line 152) medium age of patients is indicated as 51 (IQR 39-62) years while in the Table1 the same medium age 51 is indicated as (0-106) or (16-106)?
Answer: Patient ages indicated in the text are reported as median and IQR. In Table 1, age is given as median and minimum-maximum, which is reflected in the description of the variable in the "Characteristics" column.
- While the origin of the value 20 762 – the number of the analyzed operations is illustrated by Fig.1, the origin of the number of the unique patients 31 703 remains obscure.
Answer: The EUROCRINE registry recorded a total of 32008 operations performed on 31703 patients. After applying the inclusion/exclusion criteria, 20762 operations in 20691 patients remained for analysis. In Table 1, the ambiguous term 'unique patients' has been replaced with 'Total patients', as suggested by the reviewer.
Thank you once again for your constructive comments and recommendations.